# Role of Repeat Tract Structure and the rs7158733 SNP in Spinocerebellar Ataxia 3

**DOI:** 10.3390/ijms26209836

**Published:** 2025-10-10

**Authors:** Suran Nethisinghe, Hector Garcia-Moreno, Jude Alwan, Robyn Labrum, Paola Giunti

**Affiliations:** 1Ataxia Centre, Department of Clinical and Movement Neurosciences, UCL Queen Square Institute of Neurology, Queen Square, London WC1N 3BG, UK; s.nethisinghe@ucl.ac.uk (S.N.); h.garcia-moreno@ucl.ac.uk (H.G.-M.); j.alwan@ucl.ac.uk (J.A.); 2Neurogenetics Service, Rare and Inherited Disease Laboratory, London North Genomic Laboratory Hub, Great Ormond Street Hospital for Children NHS Foundation Trust, London WC1N 3BH, UK; r.labrum@nhs.net

**Keywords:** Spinocerebellar ataxia 3, SCA3, PolyQ, CAG repeat, ataxia, single-nucleotide polymorphism

## Abstract

Spinocerebellar ataxia 3 (SCA3) is a neurodegenerative condition caused by an expansion of a polyglutamine tract within the *ATXN3* gene. Normal alleles range from 12 to 44 repeats, while pathogenic alleles have 52 repeats or more. The canonical *ATXN3* repeat tract sequence includes three interruptions at positions 3 (CAA), 4 (AAG), and 6 (CAA). The intragenic rs7158733 single-nucleotide polymorphism (SNP) flanks the *ATXN3* repeat region and substitutes a TAC^1118^ tyrosine codon with a TAA^1118^ stop codon, resulting in a shorter ataxin-3aS isoform. We examined the distribution of SCA3 allele repeat sizes in a UK-based cohort presenting with an ataxic phenotype. The 6596 alleles showed a clear gap between normal and expanded alleles, with no intermediate alleles containing 41 to 57 repeats. We used clone sequencing to characterize the structure of the *ATXN3* repeat region in a sub-cohort of 44 SCA3 patients. We observed that the three canonical interruptions were typically preserved. There was no association of the interruptions with age at onset detected in this cohort, given the limited power of this sub-cohort. We genotyped the rs7158733 SNP in a sub-cohort of 79 SCA3 patients and found that 74.7% of expanded alleles carried the A^1118^ variant, which was associated with earlier disease onset. This study highlights the importance of rs7158733 genotyping alongside *ATXN3* repeat sizing for patient evaluation, as this SNP modifies the effect of repeat size on age at onset in SCA3 for pathogenic alleles up to 69 repeats.

## 1. Introduction

Spinocerebellar ataxia type 3 (SCA3), also known as Machado–Joseph disease (MJD), is an autosomal dominantly inherited polyglutamine (polyQ) disorder, representing the most common SCA worldwide. This pathology arises from a CAG repeat expansion within the *ATXN3* gene, which encodes for the protein ataxin-3. It demonstrates a spectrum of clinical presentations, including but not limited to progressive cerebellar ataxia, external ophthalmoplegia, dysarthria, dysphagia, pyramidal signs, dystonia, rigidity, and distal muscle atrophy [1]. Healthy individuals typically exhibit a range of twelve to forty-four repeats, whereas patients present with an expanded range of fifty-two to eighty-six repeats [1,2,3]. The onset of the disease spans a wide range, from four to seventy-five years [4], with the mean age of onset approximately forty years [5]. Like other polyglutamine disorders, the number of CAG repeats in the *ATXN3* expanded allele is inversely correlated with the age at which the disease manifests and directly correlates with disease severity [1,3,6,7,8].

Interruptions within the CAG repeat tract of the genes responsible for polyglutamine diseases have gained increasing attention due to their potential impact on disease severity, diagnostic accuracy, genetic counselling, elucidation of disease mechanisms, and possible future therapeutic development [9,10,11]. The canonical sequence of the *ATXN3* gene repeat tract includes three interruptions at positions 3 (CAA), 4 (AAG), and 6 (CAA) [2]. Like other repeat expansion disorders, SCA3 exhibits repeat instability in both germline and somatic cells. Interruptions within the repeat tract are believed to impact this instability, as uninterrupted stretches are typically more prone to instability and associated deleterious effects.

Three intragenic single-nucleotide polymorphisms (SNPs) flanking the *ATXN3* CAG repeat tract were originally studied in SCA3 patients, with their initial nomenclature being based on the sequence described in Kawaguchi et al. [2]. A^669^TG/G^669^TG (rs1048755, A^669^/G^669^) is located 11.4 kbp upstream from the CAG repeat in exon 8 [12,13]. C^987^GG/G^987^GG (rs12895357, C^987^/G^987^) is immediately downstream to the CAG repeat tract in exon 10 [2]. Finally, TAA^1118^/TAC^1118^ (rs7158733, A^1118^/C^1118^) is located 132 bp downstream of the CAG repeat tract within the 3′ UTR [13,14]. Here, we focus on the rs7158733 SNP where the ochre stop codon TAA^1118^ is substituted for the codon TAC^1118^. Previous studies have found the C^1118^ to be more common in chromosomes without the expansion (63% of 124 control chromosomes [13] and 75% of 303 non-expanded chromosomes [14]). Conversely, the A^1118^ allele was particularly frequent in chromosomes carrying more than 33 repeats (73%) [14]. These results were corroborated in a worldwide haplotype study, where 76% of expanded chromosomes were found to carry the A^1118^ allele, whereas this was present in only 24% of the control chromosomes [15] and similar results have been reproduced in other cohorts [16]. The pathological significance of the rs7158733 (A^1118^/C^1118^) SNP has remained elusive. Previous studies have been unable to identify an association between the A^1118^/C^1118^ alleles and age at onset in patients with SCA3/MJD [14,17], nor prove an influence of the SNP on a preliminary analysis of patients’ clinical presentation [14]. However, data regarding the effect of the SNP on clinical progression or association with other biomarkers have not yet been reported. The rs7158733 SNP seems to have an effect at both the RNA and protein levels. The presence of the TAA^1118^ codon has been predicted to have an impact on RNA secondary structure and the binding of RNA-binding proteins (RBPs) to the *ATXN3* transcript [17]. The presence of the stop codon TAA^1118^ also produces a shorter isoform of ataxin-3, referred to as ataxin-3aS (short), whilst the presence of the TAC^1118^ codon produces the ataxin-3aL (long) isoform (Figure 1). Compared to other isoforms, ataxin-3aS has a lower protein concentration and shorter half-life, resulting from its degradation through both the autophagy pathway and the ubiquitin–proteasome system (UPS). However, ataxin-3aS shows a higher nuclear localization and insolubility than other isoforms, producing larger protein aggregates than ataxin-3aL [18]. Together, these characteristics indicate that ataxin-3aS is more prone to aggregation and, therefore, is likely to have a higher pathogenicity.

In this study, we aim to determine whether genetic biomarkers, such as the repeat tract configuration in the expanded *ATXN3* allele and the rs7158733 SNP variant, are associated with distinct SCA3 phenotypes in our cohort. In particular, we seek to understand how these factors influence the age at onset, disease duration, and clinical presentation of SCA3. Additionally, we aim to identify any intermediate alleles in our large cohort, considering the unique gap between normal and pathogenic alleles observed in SCA3 that has remained unresolved since the discovery of the gene over 20 years ago [2], unlike SCA1 (Appendix A).

## 2. Results

### 2.1. ATXN3 Allele Distribution in a Large UK Cohort

A summary of the cohorts and sub-cohorts is this study is shown in Figure 2. SCA3 diagnostic tests were performed on 6596 discrete chromosomes from 3298 individuals by the Neurogenetics Unit at the National Hospital for Neurology and Neurosurgery, London. The frequency and distribution of the repeat sizes are shown in Figure 3. Normal alleles in this cohort range from 12 to 40 repeats (n = 6510; mean = 21 repeats), and are shown in Figure 3A, whilst expanded alleles that range from 58 to 91 repeats (n = 86; mean = 69 repeats) are shown in Figure 3B. The median normal allele was 22 repeats, with an interquartile range of 6 repeats. The modal allele also contained 22 repeats and was detected in 1471 chromosomes (22.3% of all tests; 22.6% of normal alleles). The median expanded allele was 70 repeats, with an interquartile range of 7 repeats. The modal allele also contained 70 repeats and was detected in 16 chromosomes (0.24% of all tests; 18.6% of expanded alleles).

There was a clear distinction between normal and expanded alleles, and no intermediate alleles of 41 to 57 repeats were observed in the cohort.

### 2.2. Clone Sequencing

A total of 440 clones were sequenced from the 44 subjects in the “Cloning” sub-cohort. There were 286 clones (65.0%) that contained expanded alleles, whilst 154 (35.0%) contained non-expanded alleles. The median number of clones per participant was 9.5 (Q1, Q3 = 8, 12). Individual CAG repeat sequences from these clones can be found in Appendix A. Since the number of clones sequenced for each individual differed, clone numbers were expressed as a percentage to account for clone depth (Appendix A).

The canonical *ATXN3* repeat tract contains three interruptions towards its 5′ end: CAA at position 3, AAG at position 4, and CAA at position 6 [2]. In the “Cloning” sub-cohort, 407 out of 440 clones (92.5%) contained sequences with the canonical *ATXN3* repeat tract. A total of 29 participants in the “Cloning” sub-cohort (65.9%) had clones with only canonical repeat tract sequences. Information about the most frequent non-expanded allele, the most frequent expanded allele, and the most frequent loss of canonical interruption allele can be found in Table 1.

#### 2.2.1. Loss of Canonical Interruptions

There were 25 clones (5.7%) containing sequences that lacked one or more canonical interruptions in expanded alleles that belonged to eight patients (18.2% of the “Cloning” sub-cohort). Four of these eight participants (Participants #8, 9, 17, and 42) showed non-canonical sequences in all clones for their expanded alleles, whilst the other four presented with a combination of canonical and non-canonical clones for their expanded alleles (Participants #5, 13, 25, and 31). None of the clones lacking one or more of the canonical interruptions contained additional downstream interruptions in the CAG tract. The canonical CAA interruption at position 6 was lost in 23 clones, whereas the AAG interruption at position 4 was missing in four clones, and the CAA interruption at position 3 was only absent in a single clone with a pure CAG stretch.

#### 2.2.2. Presence of Additional Interruptions

Eight clones contained one additional CAA interruption in the CAG repeat tract and belonged to the expanded alleles from seven subjects (15.9% of the “Cloning” sub-cohort). However, all of these participants had clones with canonical alleles and clones with an additional interruption in their expanded allele sequences. The additional interruptions arose from CAG to CAA codon transitions in positions 8 (n = 3), 9 (n = 2), 10 (n = 1), 12 (n = 1), and 25 (n = 1). None of the clones with additional interruptions showed loss of any of the canonical interruptions.

#### 2.2.3. Correlation with Age at Disease Onset

For the 44 subjects in the “Cloning” sub-cohort, age at disease onset information was available for 34 patients (77.3%). We examined the influence of pathogenic allele repeat length on the patient’s age at onset, finding a negative correlation between the age at disease onset and the median pathogenic allele repeat tract size (Figure 4), with a Pearson correlation coefficient *r* = −0.360 (*p* = 0.037) and fitting a linear regression model (slope = −1.039, y-intercept = 114.6, *R*^2^ = 0.129).

#### 2.2.4. Non-Expanded Alleles

The size of non-expanded alleles had a unimodal distribution, with alleles of 23 CAG repeats being the most frequent (n = 8; 21.6%). The median number of CAG repeats in non-expanded alleles was 23.0 repeats (Q1, Q3 = 22.0, 30.0). All clones carrying non-expanded alleles contained canonical sequences.

### 2.3. rs7158733 SNP (A^1118^/C^1118^) Allele Frequencies

For expanded alleles, 59 out of 79 alleles (74.7%) carried the A^1118^ variant of the SNP, whilst for non-expanded alleles, 26 out of 79 (32.9%) carried the A^1118^ variant of the SNP (Table 2). There was a statistically significant association between the expanded allele and the A^1118^ variant of the SNP (McNemar’s $\chi^{2}\;$= 18.46; df = 1; *p* < 0.001), with an OR = 3.54 (95% CI with exact approximation: 1.88, 7.14).

The demographic and clinical characteristics of the SCA3/MJD participants based on their rs7158733 SNP genotype are shown in Table 3. The proportion of females was similar in both groups (Table 3).

Participants with the A^1118^ allele were significantly younger at baseline compared to C^1118^ carriers (difference in medians = 6 years) (*p* = 0.029) (Table 3 and Figure 5A), with a significantly earlier age at onset of the disease (difference in medians = 14) (*p* = 0.017) (Table 3 and Figure 5B). There was no significant difference in either the disease duration (difference in medians = −2) (*p* = 0.583) (Table 3 and Figure 5C) or the size of the expanded allele (difference in medians = −2) (*p* = 0.294) (Table 3 and Figure 5D) between subjects with A^1118^ and C^1118^. There was also no statistically significant difference in the distribution of the rs7158733 SNP based on ethnicity ($\chi^{2}$= 9.88; df = 4; Fisher’s exact test, *p* = 0.054) (Appendix A).

When examining the relationship between age at onset and the number of CAG repeats in the expanded allele, stratifying based on expanded allele rs7158733 SNP (A^1118^/C^1118^) variant, we found both SNP groups had a negative correlation between their age at disease onset and pathogenic allele repeat tract size (Figure 6). The A^1118^ group (n = 48) had a Pearson correlation coefficient r = −0.320 (*p* = 0.027) and fit a linear regression model (slope = −0.779, y-intercept = 90.81, *R*^2^ = 0.102). The C^1118^ group (n = 15) had a Pearson correlation coefficient r = −0.862 (*p* < 0.0001) and fit a linear regression model (slope = −3.284, y-intercept = 264.4, *R^2^* = 0.743). There was a very significant difference in the regression line slopes (F = 9.348; *p* = 0.003), suggesting that the SNP acts as a modifier of the effect of the number of CAG repeats on the age at disease onset (Figure 6). Multiple linear regression analysis showed an earlier age of onset in the A^1118^ group adjusted by the number of CAG repeats in the expanded allele (−7.61 years on average, *p* = 0.032), with larger differences in age of onset for lower numbers of CAG repeats in the expanded allele (Figure 6). In our cohort, sex and the number of CAG repeats in the non-expanded allele were not associated to age of onset

The effect of the number of CAG repeats in the non-expanded allele on age at disease onset was also explored. However, analyses showed no association between the number of CAG repeats in the non-expanded allele and the age at onset of the disease, stratifying based on rs7158733 SNP (A^1118^/C^1118^) variant.

When examining the baseline patient rating scales, there was no significant difference in scale for the assessment and rating of ataxia (SARA), inventory of non-ataxia signs (INAS), or activities of daily living (ADL) scores between patients with the A^1118^ rs7158733 SNP and patients with the C^1118^ SNP in *cis* with their expanded allele (Appendix A). Longitudinal progression changes in SARA, INAS, and ADL were also examined, and no significant differences between the rs7158733 SNP groups were found.

Moreover, we wanted to understand whether there was an association between the presence of the rs7158733 SNP and the presentation of the disease. Appendix A summarises the symptoms at onset in the “Genotyping” sub-cohort. In both groups, the most common symptom was ataxia, with very few patients displaying parkinsonism or spasticity as initial symptoms. There was no significant difference in the symptoms at onset between groups.

## 3. Discussion

We adopted a deep-genotyping approach to improve our understanding of the genetics of SCA3. Firstly, we aimed to identify intermediate alleles in a large cohort of subjects who have undergone SCA3 tests. This would close the gap between normal and pathogenic alleles currently seen in SCA3, which has not been observed for other CAG repeat disorders, where instability in intermediate and reduced penetrance alleles leads to a repeat expanding past the pathogenic threshold (Appendix A). This study showed a prominent gap between normal and pathogenic alleles in our cohort, with no intermediate alleles detected with 41 to 57 repeats. This gap has been previously described by several groups in different populations. When *ATXN3* was first cloned in 1994, no intermediate alleles between 37 and 67 repeats were found in this Japanese cohort [2]. Subsequent studies in an Azorean cohort [19], a Taiwanese cohort [20], British cohort [3], a Dutch individual [21], and others have closed the gap to between 45 and 51. Examining the SCA3 diagnostic tests subsequently performed at the NHNN between 2015 and 2020, we found that the intermediate-allele gap is still present with no alleles between 37 and 57 repeats (774 alleles were between 14 and 36 repeats and 30 alleles were between 58 and 76 repeats). A shift to using tethered-PCR for sizing accuracy [22] as further confirmed the gap. Out of 704 SCA3 tests performed at the North Thames Genomics Laboratory Hub (GLH) between 2020 and the present date in 2025, 662 had repeat sizes of 14 to 39, whilst 42 had repeat sizes of 59 to 79. This shows the intermediate gap is still 40–58 repeats.

The confirmation of the gap between normal and pathogenic alleles in our cohort suggests that this unique behaviour for SCA3 may be due to the founder origins of the Machado and Joseph lineages, where such limited haplotypes have not been observed for other polyglutamine disorders [12,15,23,24,25,26,27,28,29,30,31,32,33]. The gap is still distinct when using an improved repeat sizing technique, spinocerebellar ataxia tethering PCR [22]. Diagnostic tests performed at the North Thames Genomics Laboratory Hub (GLH) revealed that out of 704 SCA3 diagnostic tests performed to date from 2020, no intermediate alleles were detected with 40 to 58 repeats.

To study the repeat tract structure and the role of interruptions in SCA3 pathology, we cloned and sequenced the *ATXN3* repeat tract in a sub-cohort of patients. We sequenced 440 clones from 44 patients, with the majority (286 clones, 65.0%) containing expanded alleles. A negative correlation was observed between the age at disease onset and the median pathogenic allele repeat tract size. Most clones (92.5%) contained sequences with the canonical *ATXN3* repeat tract with the three interruptions at positions 3 (CAA), 4 (AAG), and 6 (CAA), indicating that the loss or gain of any interruptions is rare. The canonical interruptions in the *ATXN3* repeat tract appear consistent across different human populations and even in some non-human primates (e.g., chimpanzees and gorillas) [34]. Unlike most CAG repeat genes, such as *ATXN1*, the interruptions in *ATXN3* are not completely lost from the repeat tract during pathological expansion. In our cohort, all expanded alleles, except one potential artefact, contained interruptions, with the polymorphic CAG present at position 7 of the tract in these alleles. Four subjects (9.1% of the “Cloning” sub-cohort) exhibited a consistent loss of the CAA interruption at position 6, a change previously described for non-expanded alleles [2]. Our data indicate that changes in the repeat tract structure are rare, and interruptions do not seem to influence the age at onset in SCA3, unlike interruptions in SCA1 [9] and Huntington’s Disease (HD) [35,36], where interruptions delay age at onset, and their loss leads to an earlier age at onset.

We genotyped a sub-cohort for the rs7158733 SNP and found that 74.7% of expanded alleles carried the A^1118^ variant of the SNP, a frequency concordant with previous studies [14,15,16]. Individuals with the A^1118^ SNP in *cis* with their expanded allele had a significantly lower baseline age and an earlier age at onset than those with the C^1118^ SNP, despite both groups having similar median expanded allele repeat sizes. Previous studies could not find differences in age at onset between the A^1118^ and C^1118^ groups [14,17], which could be explained by differences in the composition of the cohorts and/or data collection. For instance, Melo et al. studied an Azorean patient cohort with a predominance of the C^1118^ SNP in *cis* with expanded alleles (74.2%) and considered the age at onset as the “self-reported age of first symptoms of the disease” [17]. The Azorean population presents with the highest worldwide prevalence of SCA3/MJD [37]. This could bias the reported age at onset towards earlier values because of increased awareness in that population and better recognition of very early symptoms in the disease course. Additionally, European studies such as the European integrated project on spinocerebellar ataxias (EUROSCA) and the European spinocerebellar ataxia type 3/Machado–Joseph disease initiative (ESMI) protocols consider age at onset of gait difficulties as the age at onset of the disease, even if these are not the first symptoms of the disease.

Alleles carrying the TAA^1118^ codon code for a shorter ataxin-3 isoform (ataxin-3aS) [18]. Ataxin-3aS has a shorter in vitro half-life compared to ataxin-3aL and ataxin-3c, due to combined degradation via autophagy and the ubiquitin-proteasome system [18]. Additionally, ataxin-3aS shows higher enzymatic activity, a preferential nuclear localisation (even for fragments with low numbers of glutamines), and greater insolubility compared to the other isoforms [18]. These properties indicate that ataxin-3aS is a more pathogenic isoform, and hence, patients in the A^1118^ group would be expected to present with a more severe condition than C^1118^ carriers.

When stratifying the sub-cohort based on the rs7158733 (A^1118^/C^1118^) SNP, the regression line slope for the C^1118^ group is significantly steeper compared to the A^1118^ group. This indicates that for pathogenic allele sizes up to 69 repeats, the age at onset for individuals with a C^1118^ SNP is later than for those with an A^1118^ SNP. The milder phenotype in terms of age at onset within the C^1118^ group supports the proposition that the ataxin-3aS isoform expressed by the A^1118^ group is more pathogenic. For lower repeat numbers, the A^1118^ allele produces a more toxic form of ataxin-3 (ataxin-3aS), resulting in an earlier age at onset. The impact of longer polyglutamine tracts would override the effect of this isoform, thus both A^1118^ and C^1118^ groups would exhibit a similar age at onset. Therefore, it is possible that the rs7158733 (A^1118^/C^1118^) SNP acts as a disease modifier in carriers of expanded alleles with lower repeat numbers.

There was no significant difference in cross-sectional patient rating scales (SARA, INAS, or ADL) or symptoms at onset between patients in the A^1118^ group and the C^1118^ group. Similarly, the individual rates of progression in SARA total scores for both groups were not significantly different, which may be attributed to the low number of patients in the C^1118^ SNP group and subjects without a follow-up visit.

Looking at the “genotyping” sub-cohort as a whole, both A^1118^ and C^1118^ groups presented with a similar phenotype at onset, with over 90% of participants in both groups experiencing ataxia as their initial symptom. Although data collection could be biased towards cerebellar gait disorders, these figures seem to agree with other studies [38,39]. These results also agree with a previous report that could not identify any differences in disease presentation between both groups [14].

This study has some limitations. The sample size is relatively modest due to SCA3 being a rare disorder. In particular, the C^1118^ group in the “genotyping” sub-cohort is especially small compared to the A^1118^ group. Therefore, this study may have lacked enough statistical power to detect all clinically relevant effects and confounders. Comparing our cohort to the recently published European cohort of ESMI [40], the differences we observe, for instance, in the later age at onset in individuals carrying the C^1118^ SNP, may be due to differences in cohort composition, with our smaller cohort potentially amplifying any observations. However, our cohort is more ethnically diverse and is slightly older at baseline.

Integrating SNP genotyping into the diagnosis of SCA3/MJD can identify individuals suitable for allele-specific silencing therapies. Traditional diagnostic methods rely on PCR-based techniques, with further tests used if larger expansions or interruptions are suspected. Advanced PCR protocols, such as TP-PCR and tethering PCR, streamline diagnosis but do not provide intragenic haplotype details. Recent approaches include the use of haplotype analysis in diagnosis. Future advancements in bioinformatics may enable efficient diagnosis of repeat expansion disorders through Whole Genome Sequencing (WGS) in combination with ExpansionHunter [41], offering intragenic haplotype insights [22,42,43,44,45,46].

In conclusion, we found that the rs7158733 SNP modifies the effect of repeat size on age at onset in SCA3 for pathogenic alleles up to 69 repeats. Therefore, we recommend genotyping this SNP alongside *ATXN3* repeat sizing for the diagnostic assessment of patients.

## 4. Materials and Methods

### 4.1. Ethical Statement

This study was approved by the London (Queen Square) NHS Research Ethics Committee (reference 09/H0716/53; initial approval date 17 September 2009) at the National Hospital for Neurology and Neurosurgery, London and the University College London NHS Health Research Authority (reference 17/LO/0381; approval date 28 April 2017).

### 4.2. Patient Cohort

The diagnostic test cohort consists of patients with an ataxic phenotype whose blood samples were sent to the Neurogenetics Unit at The National Hospital for Neurology and Neurosurgery in London to undergo a panel of diagnostic tests for SCA1, SCA2, SCA3, SCA6, and SCA7 between November 1993 and October 2013. For this study, we selected a cohort of 83 patients diagnosed with SCA3 who were seen at the Ataxia Centre in London. A total of 79 patients underwent genotyping for the rs7158733 SNP. Forty-four patients had their *ATXN3* repeat tracts cloned and sequenced, with 40 also included in the group genotyped for the rs7158733 SNP. Four patients from the cloning sub-cohort did not undergo rs7158733 SNP genotyping because of DNA shortage from these historic samples. Clinical data were collected from patient records, and rating scales were obtained from at least one visit.

### 4.3. SCA3 Fragment Sizing and Cloning of SCA3 Allele CAG-Repeat Tracts

Genomic DNA were extracted as previously described [9]. SCA3 alleles were fragment sized by amplifying the CAG-repeat tracts with a FAM-labelled PCR primer, SCA3F (6-FAM-5′-CCAGTGACTACTTTGATT-3′), and SCA3R (5′-TGGCCTTTCACATGGATGTGAA-3′) before resolving fragments on an ABI 3730*xl* DNA analyzer with a GeneScan 500 LIZ Size Standard (Thermo Fisher Scientific, Waltham, MA, USA). Repeat lengths were calculated by subtracting the number of flanking bases in the PCR product outside the repeat region (172 bp) and then dividing by 3.

Two strategies were employed for clone sequencing. The CAG-repeat tracts were amplified with either *ATXN3*x12 *Bam*HI Forward (5′-taccgagctcggatccGTGTCAAACTTCTGACCTCAAGCC-3′) and *ATXN3*x12 *Xho*I Reverse (5′-gccctctagactcgagATGAATGGTGAGCAGGCCTTACCT-3′) primers, digested with *Bam*HI and *Xho*I restriction enzymes and cloned into a pcDNA3.1(+) vector or with *ATXN3*x12 Rpt SNP For (5′-ACCACTCCTGGCCATGATAG-3′)and *ATXN3*x12 Rpt SNP Rev (5′-AGCAATCCTCTCCTGCCTTG-3′) primers and ligated into a pCR-Blunt vector using the Zero Blunt PCR Cloning kit (Thermo Fisher Scientific). Plasmids were propagated in Stbl3 *E. coli* (Thermo Fisher Scientific).

### 4.4. Characterisation of the rs7158733 SNP (A^1118^/C^1118^)

Genotyping of the rs7158733 SNP (A^1118^/C^1118^) was performed by allele-specific PCR using a 6-FAM-labelled forward primer and an unlabelled allele-specific reverse primer as previously described [16]. The primers used were rs7158733 For (FAM-5′-CCAGTGACTACTTTGATTCG-3′) and either rs7158733 C Rev (5′-AAAAATCACATGGAGCTCG-3′) for TAC^1118^ or rs7158733 A Rev (5′-AAAAATCACATGGAGCTCT-3′) for TAA^1118^. The thermocycling conditions were 94 °C for 2 min; 10 cycles of 94 °C for 15 s, 70 °C for 15 s decreasing 1 °C per cycle, 68 °C for 15 s; 32 cycles of 94 °C for 15 s, 60 °C for 30 s, 68 °C for 15 s using Invitrogen™ Platinum™ II Hot-Start PCR Master Mix (Thermo Fisher Scientific). PCR products were purified using a GeneJET Gel Extraction and DNA Cleanup Micro Kit (Thermo Fisher Scientific) according to the manufacturer’s protocol, eluting with 10 µL elution buffer.

A total of 1 µL of purified 6-FAM-labelled PCR products was combined with 12 µL HiDi Formamide (Applied Biosystems, Waltham, MA, USA) and 0.3 µL GeneScan 500 LIZ^®^ Size Standard (Applied Biosystems) and resolved on an ABI 3730*xl* DNA Analyzer (Applied Biosystems). Fragment analysis was performed using the Peak Scanner analysis module on the Thermo Fisher Cloud. The CAG repeat length for each allele associated with an rs7158733 SNP was calculated by subtracting 268 bp from the fragment size and dividing this by 3.

### 4.5. Statistical Analysis

Statistical analysis was conducted with Prism (version 10.2.3, GraphPad Software, Inc., Boston, MA, USA) and Stata (version 17.0, StataCorp LLC, College Station, TX, USA). Graphs were prepared in Prism.

In the “Cloning” sub-cohort, for a single participant, each allele (expanded or normal) is represented by a variable number of clones. For each participant, the size of their pathogenic expanded and normal alleles was calculated as the median size of all the clones representing that given allele. The median was chosen rather than the mean to avoid the influence of extreme observations on the data. The relationship between age at onset and median pathogenic allele size was examined using Pearson’s *r* correlation coefficient, and the data were fitted to a simple linear regression model.

McNemar’s test for paired data was used to examine the association of the rs7158733 SNP variant with the expanded allele.

Groups for comparison were defined by the variant of the rs7158733 SNP in the expanded allele (A^1118^ or C^1118^). Bivariate analysis of differences in demographic, genetic variables, and disease characteristics between the groups was performed using Pearson’s $\chi^{2}$ tests and Mann–Whitney’s tests. Linear regression slopes for rs7158733 SNP groups were compared using a two-tailed F test. Adjusted effects of the A^1118^/C^1118^ SNP on age of onset were also analysed through multiple linear regression models, including the following covariates in the maximum model: number of CAG repeats in the expanded allele, and the interaction between the number of CAG repeats in the expanded allele and the SNP allele. To examine a possible effect of the number of CAG repeats in the non-expanded allele on age of onset, multiple linear regression models were fitted with the following covariates: version of the A^1118^/C^1118^ SNP, and number of CAG repeats in the expanded allele.

Data on differences in baseline patient rating scales (SARA, INAS, and ADL) were analysed with unpaired two-tailed *t*-tests.

Fisher’s exact tests were used to compare the frequency of different symptoms at onset in both groups.

A *p*-value of ≤0.05 was considered statistically significant.

## Figures and Tables

**Figure 1 ijms-26-09836-f001:**
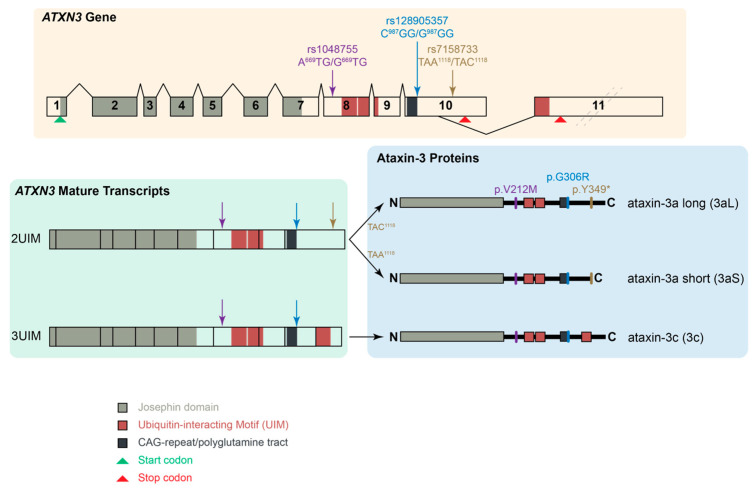
Schematic showing the *ATXN3* gene including the three intragenic SNPs flanking the CAG repeat. There are two full-length isoforms of ataxin-3, which differ based on the inclusion of a third UIM motif encoded by the differentially spliced exon 11. The rs7158733 SNP TAA^1118^ variant results in a premature stop codon and a shorter isoform, ataxin-3aS.

**Figure 2 ijms-26-09836-f002:**
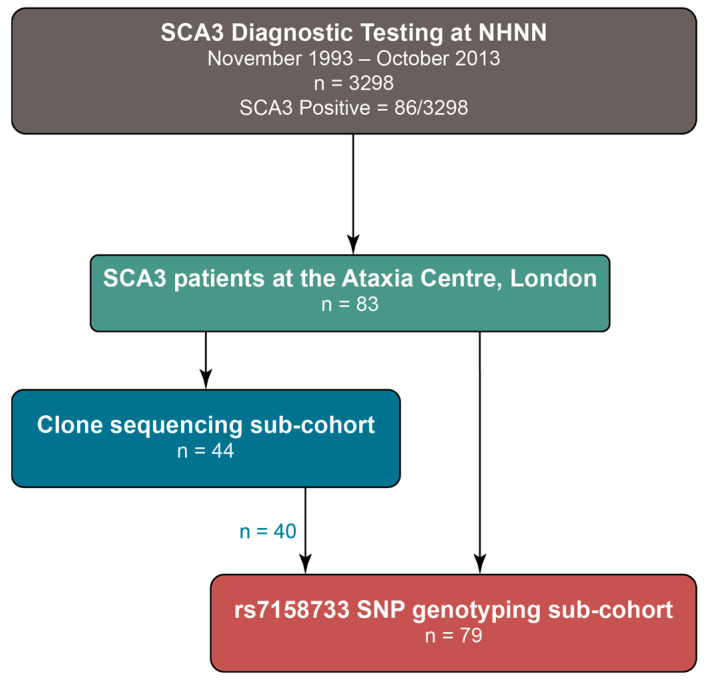
Summary of the cohorts and sub-cohorts in this study. In total, 79 SCA3 patients from the Ataxia Centre, London, underwent genotyping for the rs7158733 SNP. Of those, 44 SCA3 patients had their *ATXN3* repeat tracts cloned and sequenced; this included forty patients who were also genotyped for the rs7158733 SNP. Four patients from the cloning sub-cohort did not undergo rs7158733 SNP genotyping because of DNA shortage.

**Figure 3 ijms-26-09836-f003:**
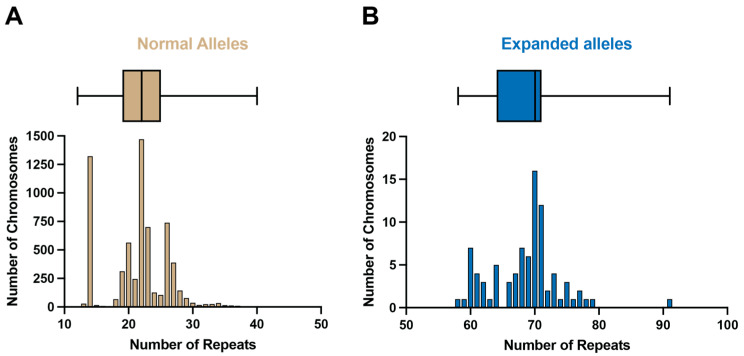
Frequency distribution of spinocerebellar ataxia type 3 (SCA3) alleles in a UK cohort at the Neurogenetics Unit, National Hospital for Neurology and Neurosurgery, London. In total, 6596 discrete chromosomes were analyzed for SCA3. Normal alleles range from 12 to 40 repeats (n = 6510; mean = 21 repeats) (**A**), whilst expanded alleles range from 58 to 91 repeats (n = 86; mean = 69 repeats) (**B**). The bar charts show the frequency distribution, whilst the box-and-whisker plots above indicate the descriptive statistics for alleles in the normal and expanded range. The box indicates the first quartile, the median, and the third quartile, and the whiskers indicate the minimum and maximum values.

**Figure 4 ijms-26-09836-f004:**
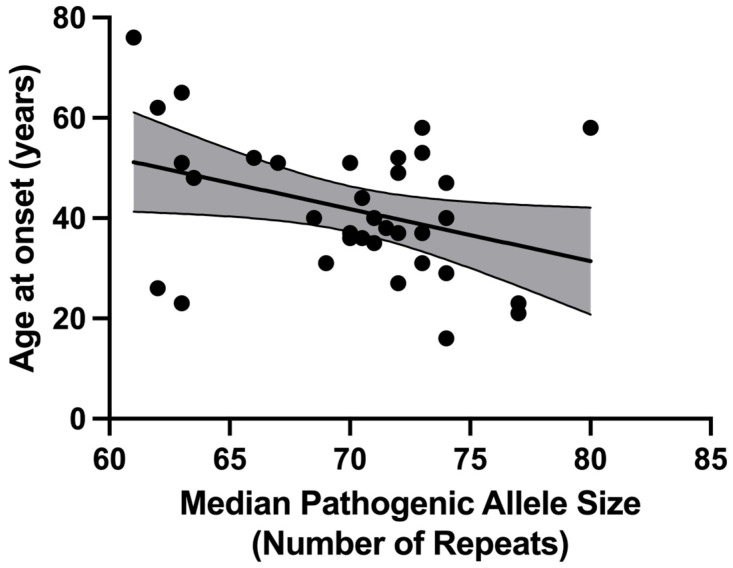
Scatter-plot showing the correlation between SCA3 pathogenic allele size and age at disease onset. Median pathogenic allele size was determined from sequenced clones, rounded to the nearest whole repeat, based on the total length of the entire CAG/CAA/AAG repeat tract. Age at onset data were available for 34 out of 44 subjects in the “Cloning” sub-cohort. Median pathogenic allele size showed a negative correlation with respect to age at onset (Pearson r = −0.360). The bold line depicts the linear model fit result, and the narrow lines show the 95% confidence interval bounds, shaded in grey.

**Figure 5 ijms-26-09836-f005:**
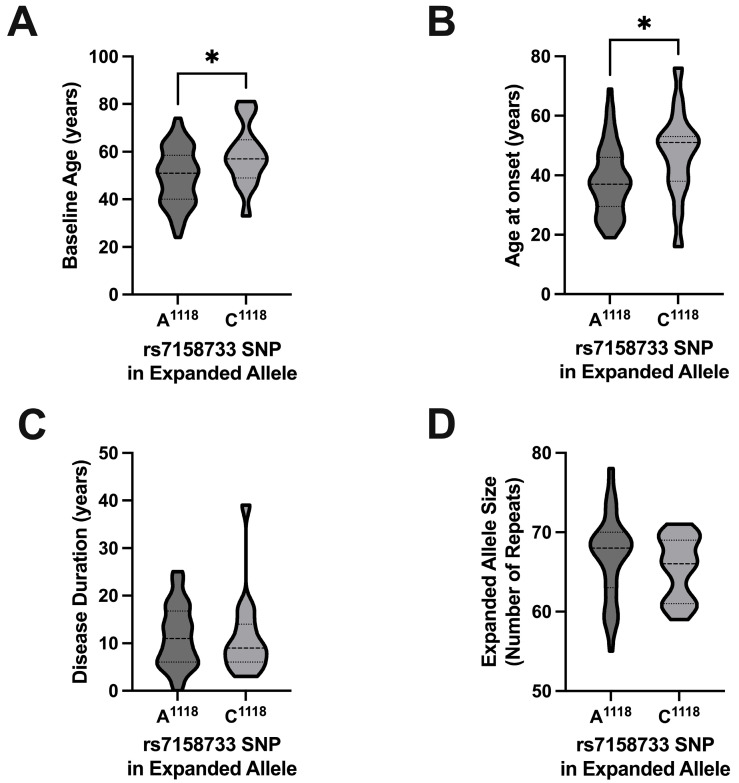
Violin plots comparing the baseline age, age at onset, and disease duration of the SCA3/MJD subjects based on their expanded allele rs7158733 SNP variant. (**A**) Participants with the A^1118^ SNP expanded allele (n = 57) were significantly younger at baseline compared to those with the C^1118^ SNP expanded allele (n = 15) (difference in medians = 6; Mann–Whitney’s test, *p* = 0.029). (**B**) Participants with the A^1118^ SNP expanded allele (n = 56) had a significantly earlier age at disease onset compared to those with the C^1118^ SNP expanded allele (n = 15) (difference in medians = 14; Mann–Whitney’s test, *p* = 0.017). (**C**) There was no significant difference in the disease duration between participants with the A^1118^ SNP expanded allele (n = 56) and the C^1118^ SNP expanded allele (n = 15) (difference in medians = −2; Mann–Whitney’s test, *p* = 0.583). (**D**) There was no significant difference in the size of the expanded allele between participants with the A^1118^ SNP expanded allele (n = 51) and the C^1118^ SNP expanded allele (n = 20) (difference in medians = −2; Mann–Whitney’s test, *p* = 0.294). * *p* ≤ 0.05.

**Figure 6 ijms-26-09836-f006:**
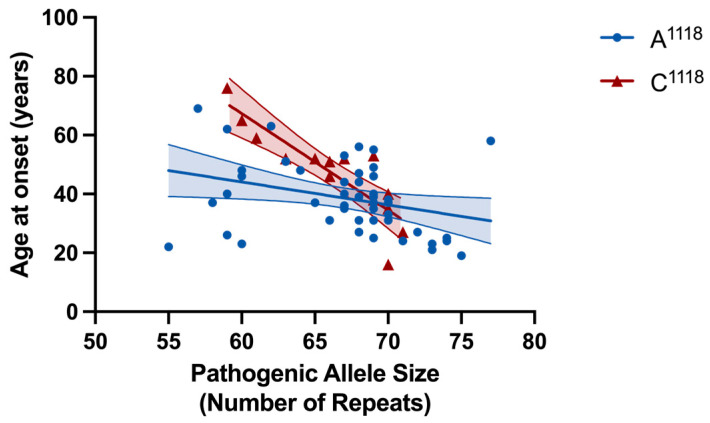
Scatter-plot showing the correlation between SCA3 pathogenic allele size and age at disease onset, stratified based on expanded allele rs7158733 SNP variant. Pathogenic allele sizes were determined from fragment sizing of rs7158733 SNP allele-specific PCR products, as detailed in the Material and Methods. Age at onset data were available for 48 out of 59 subjects with an A^1118^ SNP in *cis* with their expanded allele and 15 out of 20 subjects with a C^1118^ SNP in *cis* with their expanded allele. The blue (A^1118^) and red (C^1118^) lines depict the linear model fit result for each genotype group, and the narrow lines and shaded areas show the appropriate 95% confidence interval bounds. Both SNP groups had a negative correlation between their ages at onset and pathogenic allele sizes (A^1118^ group Pearson *r* = −0.320; C^1118^ group Pearson *r* = −0.862).

**Table 1 ijms-26-09836-t001:** Summary of clone sequencing data.

Criteria	Sequence and Information
Most frequent non-expanded allele	(CAG)_2_(CAA)(AAG)(CAG)(CAA)(CAG)_17_
23 repeats (7.5% of all clones; 10 participants)

Most frequent expanded allele	(CAG)_2_(CAA)(AAG)(CAG)(CAA)(CAG)_68_
74 repeats (5.5% of all clones; 12 participants)
Most frequent loss of canonical interruption allele	(CAG)_2_(CAA)(AAG)(CAG)_58_
62 repeats (0.7% of all clones; 1 participant, #17)

**Table 2 ijms-26-09836-t002:** Frequency of the different variants of the rs7158733 (A^1118^/C^1118^) SNP in expanded and non-expanded alleles. Percentages refer to the total number of pairs of chromosomes.

		Non-Expanded Allele		Total
		A^1118^	C^1118^	
**Expanded allele**	**A^1118^**	13	46	59 (74.7%)
	**C^1118^**	13	7	20 (25.3%)
**Total**		26 (32.9%)	53 (67.1%)	79 (100%)

**Table 3 ijms-26-09836-t003:** Demographic, genetic and disease characteristics of the SCA3/MJD subjects based on their expanded allele rs7158733 SNP variant. CAG repeat lengths were determined from fragment sizing of rs7158733 SNP allele-specific PCR products, as detailed in the Materials and Methods. Statistically significant differences are highlighted in **bold**.

	A^1118^	C^1118^	
**Female** (n = 79; %)	34 (57.6)	12 (60.0)	$\chi^{2}\;$= 0.03; df = 1; *p* = 0.852 ^a^
**Baseline age** [n = 72; years, median (Q1, Q3)]	51.0 (40.0, 58.5)	57.0 (49.0, 65.0)	***p* = 0.029** ^b^
**CAG repeats, expanded allele** [n = 71; median (Q1, Q3)]	68.0 (63.0, 70.0)	66.0 (61.0, 69.0)	*p* = 0.294 ^b^
**CAG repeats, non-expanded allele** [n = 77; median (Q1, Q3)]	22.0 (18.0, 26.0)	22.0 (20.5, 29.5)	*p* = 0.339 ^b^
**Disease duration** [n = 71; years, median (Q1, Q3)]	11.0 (6.0, 16.75)	9.0 (6.0, 14.0)	*p* = 0.583 ^b^
**Age at onset** [n = 71; years, median (Q1, Q3)]	37.0 (29.5, 46.0)	51.0 (38.0, 53.0)	***p* = 0.017** ^b^

^a^ Pearson’s $\chi^{2}$ test; ^b^ Mann–Whitney’s test.

## Data Availability

The de-identified data used in this study are available from the corresponding author.

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
