# Peer review of "Role of Repeat Tract Structure and the rs7158733 SNP in Spinocerebellar Ataxia 3"

_ijms, 2025, doi:10.3390/ijms26209836_

Round 1
Reviewer 1 Report
Comments and Suggestions for Authors
Nethisinghe et al. analyze the ATXN3 CAG repeat architecture and the flanking rs7158733 SNP (TAA1118 [A1118] / TAC1118 [C1118]) in UK-based SCA3 cohorts. Key findings are: (1) no “intermediate” alleles (41–57 repeats) between normal (12–40) and expanded (58–91) ranges in 6,596 chromosomes (Fig.2, p.3–4) ; (2) clone sequencing shows the canonical interruptions (CAA3, AAG4, CAA6) are largely preserved (92.5%) ; and (3) ~75% of expanded alleles carry A1118, which associates with lower baseline age and earlier age at onset compared with C1118 (Table 2, Fig.4–5) . The authors propose rs7158733 modifies the repeat-size/age-at-onset relationship for alleles up to ~69 repeats and recommend routine genotyping alongside repeat sizing (Abstract & Conclusion).
Major points
- Internal consistency on rs7158733’s genomic position and mechanism
In the Introduction, rs7158733 is described as “132 bp downstream … within the 3′ UTR” (exon 10) and as a variant that substitutes a stop (TAA1118) for tyrosine (TAC1118), “resulting in a shorter ataxin‑3aS isoform” (p. 2) . Please add a locus/isoform diagram clarifying how rs7158733 changes a coding stop in the relevant isoform(s). This clarification is crucial because your pathophysiological interpretation later depends on this mechanism (Discussion, pp. 8–9) . - Evidence for a true “intermediate-allele gap” needs external validation and method controls
The striking absence of 41–57-repeat alleles (Figures 1–2, pp. 3–4) is potentially practice‑changing for counseling, yet the dataset comes from a single laboratory over 1993–2013 using fragment analysis (Methods 4.2–4.3, p. 10) . Borderline sizing is vulnerable to PCR amplification bias and binning artifacts. Please strengthen this claim by:
• providing replication in a recent or multi‑centre series;
• re‑measuring a targeted 50–60 repeat window with a second technology (e.g., TP‑PCR, Southern, or PCR‑free long‑read); and
• reporting ethnicity and referral indications across time to exclude ascertainment effects (you already supply a helpful disease‑range context in Figure S3) . - Modeling of age at onset requires multivariable and non‑linear approaches
The slope difference in Figure 5 (A1118 vs C1118) is intriguing but rests on small C1118 sample size (n = 15) and a low R² in A1118 (0.102) (p. 7) . Replace or complement the bivariate fits with a model such as:
age at onset ~ expanded repeat size × rs7158733 + sex + non‑expanded allele size + baseline year/age + ethnicity + family clustering (robust/clustered SEs). - Clone‑based interruption calls and somatic mosaicism
With a median 9.5 clones per subject, clone PCR is susceptible to slippage and selection; yet conclusions about the rarity of interruption loss (5.7% of clones; eight patients) rely on these data (pp. 4–5; Tables S1–S2) . Please (i) provide per‑sample depth distributions, (ii) replicate non‑canonical findings using PCR‑free long‑read sequencing, and (iii) discuss the likelihood of somatic mosaicism vs PCR artefact, especially for the common loss of CAA at position 6 (p. 4) . - Tone down the claim that interruptions “do not influence SCA3 pathogenesis”
The Abstract states that canonical interruptions are typically preserved, “suggesting that interruptions do not influence SCA3 pathogenesis” (p. 1), but your own data mainly show that loss/gain is uncommon and not associated with age at onset in a small sub‑cohort (pp. 4–5) . Please rephrase to “no association with age at onset detected in this cohort,” and explicitly acknowledge limited power. - Clarify sample Ns and missingness across analyses
Numbers vary across sections (e.g., onset age available for 34/44 in cloning, 56–57/79 in genotype groups; Table 2 and Figures 3–5, pp. 5–7) . Expand Figure 1 (p. 3) into a CONSORT‑style flowchart that reconciles who contributed to what analyses and why data are missing (e.g., “cloning‑only” n = 4), enabling readers to track denominators at a glance .
Minor points
- Sampling rationale
Briefly justify why four patients underwent cloning only (Figure 1, p. 3); add inclusion/exclusion criteria to Methods 4.2 (p. 10) for transparency . - Supplement readability
Summarize Tables S1–S2 with a concise per‑participant digest (e.g., “most frequent sequence; % non‑canonical clones”) in the main text or a short main‑text table; currently these details are hard to analyze at a glance (Supplementary) . - Data availability
Consider depositing a minimal de‑identified dataset (repeat sizes, rs7158733, sex, ethnicity, age at onset, baseline age) in a suitable repository instead of “available upon reasonable request” (p. 12) .
Author Response
Please see attachment our response to reviewer 1

Reviewer 2 Report
Comments and Suggestions for Authors
In this manuscript, Nethisinghe and co-authors present a genetic study about SCA3. They evaluated the allele distribution (normal or expanded) of more than 3000 individuals presenting an ataxic phenotype. Furthermore, they sequenced the ATXN3 repeat region and genotyped the rs7158733 SNP, to understand the influence of these factors in the pathogenesis of SCA3. Their results showed that integrating SNP genotyping might be beneficial for the diagnosis of SCA3 and to identify individual-specific therapies. The study is well-designed, well-written and the conclusions match the results. The discussion is complete, comparing their results to other polyglutamine disorders to help the reader to understand the significance of the study.
My concerns about the manuscript are minor and just focused on improving its readability and understanding:
- Some abbreviations are explained in the text the first time that they are mentioned, whereas other are only explained in the Abbreviations section. I found this confusing, and I would choose one (preferably explaining them the first time they appear) to make it easier for the reader to find the definitions.
- Line 150-151 contain a sentence that I find confusing/unfinished, please clarify the meaning or re-word it.
- Table 2 is missing one significant difference highlighted in bold (age of onset).
The main question is to understand the importance of the structure of ATXN3 and the number of CAG repetitions in the pathophysiology of the SCA3 disease. The authors address this question in a population with ethnic variability, which is novel and introduces new relationships between the SNP and the age of onset of the disease (never described before). The methodology is simple and well-designed, and their conclusions match the results (although they also describe the limitations of this study and, thus, the limitations of the conclusions). The references are appropriate, with around 6% of self-citations that are needed for the proper understanding of the manuscript. Tables are descriptive and informative, and minor details were corrected in the previous review.
Author Response
Please see the attachment that contained our response for the Reviewer 2 comments
